# The Co-Occurrence of T-2 Toxin, Deoxynivalenol, and Fumonisin B1 Activated the Glutathione Redox System in the EU-Limiting Doses in Laying Hens

**DOI:** 10.3390/toxins15050305

**Published:** 2023-04-23

**Authors:** Szabina Kulcsár, Benjámin Kövesi, Krisztián Balogh, Erika Zándoki, Zsolt Ancsin, Márta Erdélyi, Miklós Mézes

**Affiliations:** 1Department of Feed Safety, Institute of Physiology and Nutrition, Hungarian University of Agriculture and Life Sciences, Szent István Campus, 2100 Gödöllő, Hungary; szabina.kulcsar@gmail.com (S.K.);; 2ELKH-MATE Mycotoxins in the Food Chain Research Group, Hungarian University of Agriculture and Life Sciences, 7400 Kaposvár, Hungary

**Keywords:** T-2 toxin, deoxynivalenol, fumonisin B_1_, oxidative stress, glutathione redox system, lipid peroxidation, laying hen

## Abstract

Different mycotoxins in feed lead to combined exposure, increasing adverse effects on animal health. Trichothecene mycotoxins have been associated with inducing oxidative stress, which is neutralized by the glutathione system within the antioxidant defense, depending on the dose and duration of exposure. T-2 toxin, deoxynivalenol (DON), and fumonisin B1 (FB1) are commonly found in feed commodities simultaneously. In the present study, the intracellular biochemical and gene expression changes were investigated in the case of multi-mycotoxin exposure, focusing on certain elements of the glutathione redox system. In a short-term feeding trial, an in vivo study was performed with low (EU-proposed) doses: T-2/HT-2 toxin: 0.25 mg; DON/2-AcDON/15-AcDON.: 5 mg; FB1: 20 mg/kg feed, and high doses (twice the low dose) in laying hens. The multi-mycotoxin exposure affected the glutathione system; GSH concentration and GPx activity was higher in the liver in the low-dose group on day 1 compared to the control. Furthermore, the gene expression of antioxidant enzymes increased significantly on day 1 in both exposure levels compared to the control. The results suggest that when EU-limiting doses are applied, individual mycotoxins may have a synergistic effect in the induction of oxidative stress.

## 1. Introduction

The co-occurrence of mycotoxins in poultry feeds is a primary problem worldwide; thus, the evaluation of their combined effects is essential. Corn is an important component of poultry feed commonly contaminated with *Fusarium* molds, thus potentially exposing laying hens to trichothecene mycotoxins, T-2 toxin, HT-2 toxin, nivalenol, deoxynivalenol (DON), and other fusariotoxins, such as fumonisins [1]. Many mycotoxins are known in our environment, which can be carcinogenic, hepatotoxic, and nephrotoxic in humans and animals [2]. Trichothecene mycotoxins and fumonisins can induce subchronic and subclinical symptoms in farm animals, such as growth retardation, feed refusal, impaired production traits, increased mortality, and increased sensitivity to infectious diseases due to improper immune responses and less effective vaccination [3]. However, clinical symptoms can manifest in feed refusal, vomiting, diarrhea, lethargy, reduced immune response, and low fertility [3,4].

T-2 toxin and its toxic metabolite HT-2 toxin are mycotoxins produced by *Fusarium* species in arable fields, which can cause significant damage in monogastric farm animals, even in temperate climates. In poultry, the adverse effects usually manifest in lower body weight gain, the deterioration of feed intake, lower egg production, the reduction of protein synthesis, and a weakening of the immune and antioxidant defense system [5].

DON and its active metabolites 3-acetyl DON (3-AcDON) and 15-acetyl DON (15-AcDON) are also trichothecene fusariotoxins, which can accumulate in high amounts primarily in cereals (e.g., wheat, barley, rye, and oats). Compared to other farm animal species, poultry is less sensitive to DON [6], but it can frequently be found in poultry feed, inducing long-term effects [7]. Chronic exposure causes the effect of reduced feed intake, decreased egg production, and increased lipid peroxidation. Furthermore, due to its mitigating effect, it affects the immune system, including the synthesis of immunoglobulins [8]. In addition, their neurotoxic [9], dermatotoxic [10], and emetic [11] effects were also demonstrated.

Fumonisin B1 (FB1) is the main representative form of fumonisins, which is known for damaging the immune system and causing immunosuppression in farm animals and humans [12]. Poultry is relatively resistant to FB1, but in broiler chickens and more markedly in turkeys, it can cause poor performance and increased relative organ weights and hepatitis [13]. Furthermore, FB1 is a structural analog of sphinganine, thus inducing a disturbance in the biosynthesis of sphingolipids, indirectly provoking apoptosis [14].

Trichothecenes, and other frequent mycotoxins, such as fumonisins and aflatoxins, are well known for increasing reactive oxygen species (ROS) formation by inflammatory processes and activating the xenobiotic-transforming enzyme system (e.g., the cytochrome P450 family). In addition, inflammatory processes can induce immune reactions, consequently increasing ROS formation [15,16,17]. In broiler chickens, there are a lot of data on trichothecene mycotoxin-induced oxidative damage, such as lipid peroxidation after subchronic exposure [18,19]. Based on the hierarchical model of oxidative stress, depending on its degree, mild oxidative stress activates the biological antioxidant system, including the glutathione redox system, and the expression of its encoding genes [20].

The co-occurrence of mycotoxins can cause different toxicities, enhancing or inhibiting each other’s effect. The interaction of mycotoxins can be additive, in which case the effect can be calculated as the sum of the effect of the tested toxins individually; it can be synergistic when the effect is higher than the individual ones, and antagonistic when the effect of the combination is lower than the individual [21]. Short- (subacute) and long-term (subchronic) studies have recently investigated interactions between mycotoxins. In previous research with broiler chicken [22], T-2 toxin and DON were applied together at the EU-limiting dose in a subchronic study. The results revealed that the antioxidant system was activated, which caused a lower rate of lipid peroxidation in broiler chickens. In another study with rats, [23] reduced glutathione (GSH) content and glutathione peroxidase (GPx) activity was increased by the combined effect of FB1, DON, and zearalenone (ZEA), even at a low dose of acute exposure [23], which may indicate an antagonistic effect. However, limited data are available on the toxic effect between individual trichothecene mycotoxins in the case of co-occurrence, including their action on lipid peroxidation processes and the amount or activity of the antioxidant defense, including the glutathione redox system.

The present research aimed to investigate the combined effect of short-term (72 h) exposure of the three common *Fusarium* mycotoxins, T-2 toxin and its equally toxic metabolite, HT-2 toxin, DON, and its toxic metabolites, 3-AcDON and 15-AcDON, and FB1, on laying hens. The individual EU-limiting doses of the mycotoxins were used in the low-dose group, which can frequently occur under natural conditions, and twice this amount in the high-dose group. During our measurements, some biochemical parameters of the glutathione redox system, lipid peroxidation, and expression of genes encoding the glutathione redox system were determined.

## 2. Results

### 2.1. Results of the Lipid Peroxidation Parameters

No mortality was observed in the experiment. The parameters of the initial phase of lipid peroxidation, conjugated diene (CD), and conjugated triene (CT) levels showed no significant differences at the different sampling times (Table 1). However, the marker of the termination phase of lipid peroxidation, the amount of thiobarbituric acid-reactive substances (TBARS) expressed as malondialdehyde (MDA) concentration, was significantly (*p* < 0.05) higher as an effect of low-dose exposure than the control. However, compared to the high-dose group, a significant difference was only observed on the first day of the mycotoxin exposure due to the high individual variance in all treatment groups (Figure 1).

### 2.2. Results of the Glutathione Redox System Parameters

The amount of reduced glutathione (GSH) and the activity of glutathione peroxidase (GPx) was significantly (*p* < 0.05) higher in the low-dose group than in the control and the high-dose group only on day 1 (Figure 2).

### 2.3. Relative Expression of GPX4, GSS, and GSR Genes

The overexpression of the *GPX4*, *GSS*, and *GSR* genes was found on day 1 in both the low and high mix dose groups compared to the control, and the difference between the means was statistically significant (*p* < 0.01). *GPX4* expression showed a further increase (*p* < 0.05) in the high mix group on day 2 compared to the control and low mix groups, while there was a significant (*p* < 0.01) increase in the low mix group and a decrease (*p* < 0.0001) in the high mix group than in the control by the end of the experiment (day 3). *GSS* expression was significantly elevated in the low and high mix groups (*p* < 0.05) compared to the control on day 1 and significantly (*p* < 0.001) raised on day 3 in the low mix group compared to the control and high mix groups. *GSR* expression was significantly (*p* < 0.01) higher on day 1 in the low mix and high mix groups compared to the control. By day 3, the relative gene expression only lifted significantly (*p* < 0.01) in the low mix group, while in the control and high mix groups, it did not (Figure 3).

## 3. Discussion

The biochemical results of multi-mycotoxin exposure revealed that the lipid peroxidation was in the terminal phase, as supported by the higher amount of TBARS expressed as MDA concentration in the low mix group treated with the EU-limiting dose of T-2/HT-2 toxin, DON/3-AcON/15-AcDON, and FB1 on day 1 of the experiment. After that, this lipid peroxidation parameter decreased to the control level, which suggested an early activation of the antioxidant defense, effectively inhibiting further lipid peroxidation. This result suggests early ROS formation and lipid peroxidation as an effect of low, but not high, doses of these mycotoxins. This dose-dependent difference can be explained by the phenomenon that lower amounts of mycotoxins absorb and reach the liver more rapidly than higher doses. Therefore, the early ROS formation induced the peroxidation of lipids. In addition, the ROS formation and lipid peroxide activated the antioxidant glutathione redox system, which was supported by the increased amount of GSH and activity of the GPx in the low mix groups compared to the high mix group control in the same period. The not-dose-related changes of the glutathione redox parameters support our hypothesis that there is a dose-dependent difference in the rate of absorption of mycotoxins manifested in the liver. Furthermore, these changes were observed in the laying hens fed with not the high, but the low mycotoxin-contaminated feed (EU-limiting dose applied), suggesting that there may be a synergistic effect between individual mycotoxins in terms of the activation of the antioxidant (glutathione) system. This is supported by the results of our previous research with these mycotoxins applied individually, where at the high (twice the EU-limiting) dose of T-2/HT-2 toxin, DON/3-AcDON/15-AcDON and FB1 did not increase the amount (GSH) or activity (GPx) of the glutathione redox system [24]. The relative expression of the *GPX4*, *GSS*, and *GSR* genes showed a significant increase on day 1 in both the low and high mix group compared to the control. This result suggested that the *Fusarium* multi-mycotoxin exposure induced redox changes in the cells by ROS formation, which activated the expression of antioxidant genes. However, the overexpression of these genes did not manifest at GSH and GPx levels in the case of the high-dose mix. This result suggested a time shift between the overexpression of genes and their effect on the synthesis of GSH and GPx. This result supported our hypothesis that a low dose of mycotoxins absorbs and reaches the liver earlier than a high dose, probably due to the oversaturation of membrane channels and the effect of a high dose manifested only at the gene expression level on day 1. After that, the xenobiotic detoxification was likely activated, which partly recovered the redox state of cells on day 2 of the experiment, supported by the insignificant (*GSS*) and even lower expression (*GSR*). However, *GPX4* overexpression was found even on day 2 at the high dose, suggesting that cell redox changes remain due to higher mycotoxin exposure. However, it does reach the critical level for activating GSH and GPx synthesis. The overexpression of genes of the antioxidant enzymes increased by day 3 due to low-dose mycotoxin exposure, which may be caused by the metabolism of mycotoxins, resulting in further ROS formation. These results are different from the previous results, where the relative gene expression of *GPX4*, *GSS*, and *GSR* did not change or decrease on day 1 as an effect of the high-dose exposure to the same individual mycotoxins [24]. Therefore, it can be assumed that during the 3-day multi-mycotoxin exposure, there was a synergistic effect among the *Fusarium* mycotoxins on activating the antioxidant enzymes at the gene expression level. The individual in vivo effects of these mycotoxins on the induction of oxidative stress were described earlier in broiler chickens. However, the doses applied were higher than the EU-proposed limits. The results of previous studies suggested that T-2 toxin or DON exposure in vivo increased the ROS formation in the long-term and short-term experiments [25], which activated the antioxidant defense at both the gene and protein expression levels. In addition, subacute exposure to FB1 increased the hepatic TBARS levels, inducing oxidative stress in the liver by inhibiting the ceramide synthase [26]. However, there are some studies about combined exposure with EU-limiting doses. The combined low-dose subchronic exposure of T-2 toxin and DON increased the amount or activity of the glutathione redox system and the expression of related genes. However, at the same time, there was no significantly higher rate of lipid peroxidation [22], which suggested a proper antioxidant defense. Furthermore, the common contamination of FB1 and DON of broiler feed at concentrations of the EU-recommended levels caused higher ROS levels than FB1 applied alone [27], suggesting a synergistic effect. These results suggested synergistic toxicity interactions, including changes in the redox state of cells and the activation of the antioxidant defense as an effect of multi-mycotoxin exposure. In addition, the most changes in the antioxidant markers were found when using the EU-limit values, presumably due to the development of oxidative stress even at low concentrations of mycotoxins, possibly due to their more rapid absorption from the intestine. Therefore, it is important to continue studying mycotoxin mixtures even at low concentrations and the related mechanism to better understand mycotoxin exposure-related cellular events and mycotoxin-induced diseases.

## 4. Conclusions

In conclusion, the 3-day *Fusarium* multi-mycotoxin exposure led to ROS formation, resulting in lipid peroxidation and, as a response, activating the antioxidant system. However, this effect manifested at the gene expression level, but did not affect the biochemical parameters. Furthermore, the combined effect revealed that the low dose had a more marked early effect than the high dose on lipid peroxidation and the glutathione redox system in the liver of laying hens, which was manifested in the relative expression of glutathione redox system-related genes at low and high doses.

## 5. Materials and Methods

### 5.1. Experimental design

In total, 60 Tetra SL laying hens (49 weeks of age, 90% average daily egg production) were used as an in vivo model. Three treatment groups (control, low, and high mixture groups) with 18 animals per group were formed, and six animals were used as an absolute control on day 0 of the experiment. Feed and drinking water were provided ad libitum. The nutrient content of the laying hens’ diet was 89.20% dry matter, 16.10% crude protein, 2.50% ether extract, 5.50% crude fiber, 0.79% lysine, 0.38% methionine, 0.71% methionine + cysteine, 4.12% calcium, 0.48% phosphorus (available), 0.17% sodium, 11.97 MJ/kg M. The animals were kept on deep litter with a natural light regimen (12 L/12 D). After 12 h of feed deprivation, a 3-day feeding trial was started with low- (T-2/HT-toxin: 0.25 mg; DON/3-AcDON/15-AcDON: 5 mg; FB1: 20 mg/kg feed) and high-dose (T-2 /HT-2toxin: 0.5 mg; DON/3-AcDON/15-AcDON: 10 mg; FB1: 40 mg/kg feed) multi-mycotoxin exposure. The applied EU-limiting mycotoxin contamination level of the feed [28] in the low-dose group can show casual contamination in a temperate climate [29]. In contrast, the high dose induced measurable changes in a short-term experiment. At 24, 48, and 72 h after the start of the mycotoxin exposure, post mortem liver samples were randomly collected from six animals per group, washed with isotonic saline, and stored at −70 °C until analysis.

### 5.2. Mycotoxin production

Broiler chicken feed was experimentally contaminated with T-2 toxin, DON, and FB1. T-2 toxin was produced by *Fusarium sporotrichioides* (NRRL 3299), DON by *Fusarium graminearum* (NRRL 5883), and FB1 by *Fusarium verticillioides* (MRC 826) strains on corn substrate according to the method of Fodor et al. [30]. The fungal strains were verified for the production of the mentioned mycotoxins, but the final culture was not examined for other mycotoxins. The experimental mycotoxins and their metabolites (T-2/HT-2 toxin [31], DON, 3-AcDON, 15-AcDON [32], and FB1 [33]) were measured in triplicate by HPLC equipped with a fluorescence detector after immunoaffinity cleanup. It was made with a double extraction procedure with water and methanol and by Myco6in1+® multi-mycotoxin immunoaffinity column (Vicam, Milford), as proposed by the manufacturer. The concentration of T-2/HT-2 toxin, DON/3-AcDON/15-AcDON, and FB1 was lower than LOQ in the control feed. Table 2 shows the measured mycotoxin content of the feeds used in the experiment.

### 5.3. Biochemical and Gene Expression Analyses

Initial phase markers of lipid peroxidation, CD, and CT content were determined according to the AOAC method [34] by measuring the absorption at 232 nm and 268 nm, respectively. TBARS, as final products of polyunsaturated fatty acid peroxidation, were measured in a 1:9 homogenate (in physiological saline) of liver samples using the method of Botsoglou et al. [35] and expressed as malondialdehyde, which served as a standard (1,1,3,3-tetraethoxypropane, Fluka, Buchs, Switzerland). The amount of total non-protein sulfhydryl groups expressed as GSH [36], and the activity of GPx [37], were determined in the 10.000 g supernatant fraction of 1:9 homogenate of the liver samples. GSH content and GPx activity were calculated to the protein content of the supernatant fraction using the Folin–Ciocalteu phenol reagent [38].

The gene expression measurements were performed by quantitative real-time PCR using the duplex qPCR method. From the liver, 6–10 mg samples were taken from the distal part of the right lobe, and the total RNA was purified with NucleoZOL reagent (Macherey-Nagel, Düren, Germany), according to the manufacturer’s instructions. DNase I (Thermo Fisher Scientific, San Jose, CA, USA) treatment was used to remove the genomic DNA from the purified RNA samples. The concentration, integrity, and quality of the total RNA were verified by agarose gel electrophoresis and NanoPhotometer (Implen, Munich, Germany). The purity of the RNA was accepted with the absorption OD 260/280 index ratio above 2.0. cDNA was produced with reverse transcriptase (Thermo Fisher Scientific, San Jose, CA, USA) and random nanomer primer from 1 μg of total RNA, according to the recommended protocol. In the real-time PCR measurement, pooled samples from equal amounts of cDNA per six individual animal specimens for each sampling point per treatment with five technical replicates were used, as proposed by Kendziorski et al. [39]. According to previous studies, the target genes were *GPX4*, *GSS,* and *GSR.* The endogenous housekeeping control gene was *GAPDH*, which has no interaction with oxidative stress or mycotoxins. Primers and probes were made using Primer Express 3.0.1 (Thermo Fisher Scientific, San Jose, CA, USA) software (Table 3). The real-time PCR measurements were carried out in duplexes (endogenous and target gene) using MGB-NFQ TaqMan probes (Thermo Fisher Scientific, San Jose, CA, USA) labeled with fluorescent dyes, VIC, and FAM signals (Table 4) for the simultaneous analysis of two gene products.

The relative expression level of the target genes was measured with Step One Plus™ Real-Time PCR systems (Thermo Fisher Scientific, San Jose, CA, USA) using the comparative Ct method, as described previously [24]. The VIC and FAM signals were detected at 72 °C at the end of the extension period. The ΔCt, ΔΔCt, and relative quantification (RQ = 2^-ΔΔCt^) values were calculated according to Livak and Schmittgen [40].

### 5.4. Statistical Analysis

GraphPad Prism 6.07 software (GraphPad Software, San Diego, CA, USA) was used for the statistical analysis. Data are expressed as mean ± standard deviation (SD). The normality and the homogeneity of variance were verified with the Kolmogorov–Smirnov test and Bartlett test, respectively. The data passing both tests were analyzed by one-way ANOVA and Tukey’s post hoc test (*p* < 0.05).

### 5.5. Ethical Issues

The guidelines set by the European Communities Council Directive (86/609 EEC) were followed during the experiment. The Food Chain Safety, Land Use, Plant and Soil Protection and Forestry Directorate of the Pest County Governmental Office (PE/EA/1964-7/2017) approved the experimental protocol with the lowest number of animals possible for an accurate statistical analysis.

## Figures and Tables

**Figure 1 toxins-15-00305-f001:**
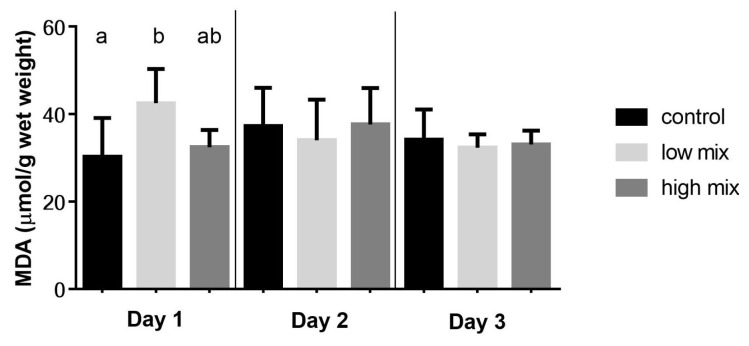
Effect of T-2/HT-2 toxin, DON/3-AcDON/15-AcDON, and fumonisin B1 co-exposure on the amount (nmol/g) of malondialdehyde of liver homogenates collected on the day 1, 2, and 3 of the trial (mean ± SD.; *n* = 6).^a, b^: different superscripts mean significant differences (*p* < 0.05). Low mix: T-2/HT-2 toxin: 0.25 mg; DON/3-AcDON/15-AcDON: 5 mg; FB1: 20 mg/kg feed. High mix: T-2/HT-2 toxin: 0.5 mg; DON/3-AcDON/15-AcDON: 10 mg; FB1: 40 mg/kg feed.

**Figure 2 toxins-15-00305-f002:**
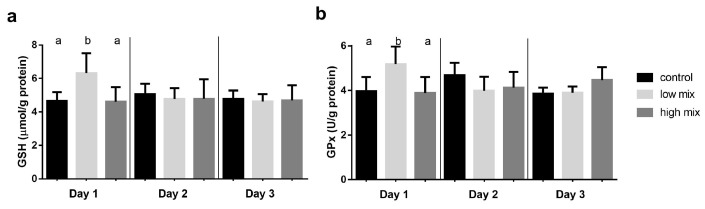
Evaluation of the antioxidant capacity: (**a**) GSH content and (**b**) GPx activity in the liver homogenates (mean ± S.D.; *n* = 6). ^a, b^: means with different letters in the same column differ significantly (*p* < 0.05). Low mix: T-2/HT-2 toxin: 0.25 mg; DON/3-AcDON/15-AcDON: 5 mg; FB1: 20 mg/kg feed. High mix: T-2/HT-2 toxin: 0.5 mg; DON/3-AcDON/15-AcDON: 10 mg; FB1: 40 mg/kg feed.

**Figure 3 toxins-15-00305-f003:**
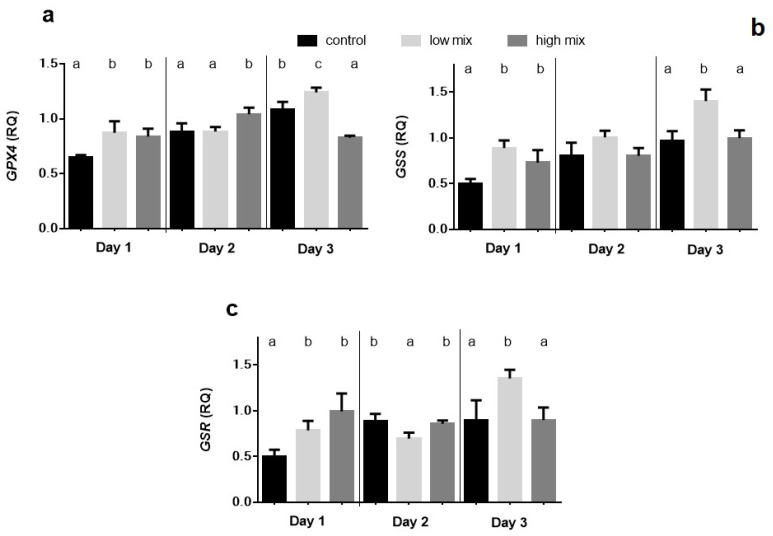
Relative gene expression of detoxification enzymes—(**a**) *GPX4*, (**b**) *GSS*, and (**c**) *GSR*—in the liver of laying hens (mean ± SD.; *n* = 6; equal amounts of cDNA per animal). RQ: relative quantification (RQ = 2^-ΔΔCt^).^a, b, c^: means with different letters in the same column differ significantly (*p* < 0.05). Low mix: T-2/HT-2 toxin: 0.25 mg; DON/3-AcDON/15-AcDON: 5 mg; FB1: 20 mg/kg feed. High mix: T-2/HT-2 toxin: 0.5 mg; DON/3-AcDON/15-AcDON: 10 mg; FB1: 40 mg/kg feed.

**Table 1 toxins-15-00305-t001:** Combined effect of short-term T-2/HT-2 toxin, DON/3-AcDON/15-AcDON, and FB1 exposure on lipid peroxidation parameters in the liver of laying hens (mean ± SD; *n* = 6).

**Conjugated Dienes (OD 232 nm)**				
	**Day 0**	**Day 1**	**Day 2**	**Day 3**
Control	0.61 ± 0.03	0.60 ± 0.07	0.60 ± 0.05	0.65 ± 0.07
Low mix		0.60 ± 0.05	0.63 ± 0.13	0.57 ± 0.04
High mix		0.60 ± 0.07	0.61 ± 0.09	0.59 ± 0.05
**Conjugated trienes (OD 268 nm)**				
	**Day 0**	**Day 1**	**Day 2**	**Day 3**
Control	0.23 ± 0.03	0.22 ± 0.02	0.22 ± 0.02	0.23 ± 0.02
Low mix		0.21 ± 0.02	0.23 ± 0.05	0.20 ± 0.02
High mix		0.22 ± 0.02	0.22 ± 0.03	0.22 ± 0.02

**Table 2 toxins-15-00305-t002:** Mycotoxin feed content used in the experiment (mg/kg feed).

Group	T-2/HT-2	DON/3-AcDON/15-AcDON	FB1
Control	<0.02	<0.02	<0.02
Low mix	0.22/0.04	4.25	21.51
High mix	0.62/0.08	11.02	39.05

**Table 3 toxins-15-00305-t003:** Primers of target and endogenous control genes.

Gene	Forward (5′-3′)	Reverse (5′-3′)	GenBank Accession No.
*GAPDH*	TGACCTGCCGTCTGGAGAAA	TGTGTATCCTAGGATGCCCTTCAG	NM_204305.1
*GPX4*	AGTGCCATCAAGTGGAACTTCAC	TTCAAGGCAGGCCGTCAT	NM_001346448.1
*GSS*	GTACTCACTGGATGTGGGTGAAGA	CGGCTCGATCTTGTCCATCAG	XM_425692.6
*GSR*	CCACCAGAAAGGGGATCTACG	ACAGAGATGGCTTCATCTTCAGTG	XM_015276627.2

**Table 4 toxins-15-00305-t004:** MGB-NFQ dual-labeled probes of genes.

Gene	MGM Dual-Labeled Probe	Fluorescent Dye
*GAPDH*	CCAGCCAAGTATGATGAT	VIC
*GPX4*	CAGCCCAATGGAG	FAM
*GSS*	AGGAGGGAACAACCTG	FAM
*GSR*	CTGGCACTTCGGCTC	FAM

## Data Availability

The raw data supporting the conclusions of this manuscript will be made available by the authors, without undue reservation, to any qualified researcher.

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
