# Peer review of "The Co-Occurrence of T-2 Toxin, Deoxynivalenol, and Fumonisin B1 Activated the Glutathione Redox System in the EU-Limiting Doses in Laying Hens"

_toxins, 2023, doi:10.3390/toxins15050305_

Round 1
Reviewer 1 Report
Dear Authors, I accept your manuscript for publication in its current form
Author Response
Reviewer 1: Dear Authors, I accept your manuscript for publication in its current form.
Response: Thank you very much for your comment.
Reviewer 2 Report
The authors have written the manuscript titled “The co-occurrence of T-2 toxin, deoxynivalenol and fumonisin B1 activated the glutathione redox system in the EU-recommended doses in laying hens” very well. I have few minor suggestions regarding the manuscript.
· In introduction part, in the last paragraph the where author measured various parameter should be discussed in the material and method part not here.
· All results are presented in tabulated form. I should recommend that few of the tables should be presented in graphical form for better understanding of the results.
· Discussion part should be written in continuous form instead of various paragraphs.
· Literature should be supported by latest references instead of old ones.
Reviewer 3 Report
Mycotoxins are toxic secondary metabolites produced by fungi, which enter the food chain through contaminated crops with which food and feed are produced. Contamination can occur in the field and/or during storage. Sometimes mycotoxins can be transferred to animals through feed and from these to products of animal origin.
During the consumption of contaminated food, mycotoxins can cause serious adverse health effects in humans, such as immunotoxic, neurotoxic or even carcinogenic effects. The combination of consumption surveys and occurrence data through total diet studies or single toxin meta-analysis approaches are mainly used to estimate dietary exposure and its effects. Results that can be significantly distorted because of differences in consumption patterns or assuming homogeneous contamination within the same food category, or because of the synergistic effect of multiple toxins.
At the metabolic level, the effects of peroxidation and metabolism of the glutathione redox system are well known, as the authors who cited Kovesi and Kulcsar are well aware. Kulcsar et al. suggested that the mycotoxins investigated individually differently activated the antioxidant defense with low-level oxidative stress at the dose applied.
The authors evaluated the overall effect of three mycotoxins, T-2 toxin, deoxynivalenol (DON), and fumonisin B1.
The results show that the final metabolic effects (peroxidation) resulting from the short duration of action (1 - 3 days) of minimal amounts of toxins (within the limits set by EU laws) were overall different from what could be expected from test on a single toxin. The high primary peroxidative effect is then decreased because of the action of the redox system. The activity of reduced glutathione and glutathione peroxidase also confirms this analysis.
The authors explain the dose-dependent difference to a more rapid hepatic absorption kinetics because of the lower blood concentration of these compounds.
The differentiated analysis of the expression of the genes coding for the antioxidant enzyme GPX4 and of the enzymes involved in the synthesis and reduction of GSH (GSS and GSR) shows a lower activity in the controls and in the group at the maximum dosage, confirming this hypothesis. Even though hens fed the high blend sometimes had increased gene expression data, they were still relatively lower than the lowest dose.
The authors highlighted some metabolic differences because of a relative synergism at low doses of the three toxins analyzed together.
Table 3 needs to be tidied up, and the data aligned.
Author Response
Reviewer 3: Table 3 needs to be tidied up, and the data aligned.
Response: The Table 3 has been replaced by diagrams.
Reviewer 4 Report
The manuscript investigated influence of co-exposure of three mycotoxins, T-2 toxin, deoxynivalenol and fumonisin B1, on the intracellular biochemical and gene expression changes of the glutathione redox system in laying hens. The study designed well and obtained some interesting results. However, some comments need to be addressed before the manuscript can be accepted:
1. In the Title: “EU-recommended doses”, I consider the word “recommended” is inappropriate, it would be better to replace it with another word, such as “limiting”.
Fumonisins was classified as one kind of trichothecene mycotoxins in this manuscript as elucidated in lines 29-32. But fumonisins does not belong to this type of fungal toxin.
2. There are many types of fungal toxins with varying degrees of toxicity, such as aflatoxins, which possess hepatotoxic activity, nephrotoxicity. I consider that it is important and necessary to briefly introduce these mycotoxin detrimental properties, and allow me to suggest the following publication to be cited herein:
Wang, Y.; Liu, F.; Zhou, X.; Liu, M.; Zang, H.; Liu, X.; Shan, A.; Feng, X. Alleviation of Oral Exposure to Aflatoxin B1-Induced Renal Dysfunction, Oxidative Stress, and Cell Apoptosis in Mice Kidney by Curcumin. Antioxidants 2022, 11, 1082. https://doi.org/10.3390/antiox11061082
Kumar, P.; Gupta, A.; Mahato, D.K.; Pandhi, S.; Pandey, A.K.; Kargwal, R.; Mishra, S.; Suhag, R.; Sharma, N.; Saurabh, V.; Paul, V.; Kumar, M.; Selvakumar, R.; Gamlath, S.; Kamle, M.; Enshasy, H.A.E.; Mokhtar, J.A.; Harakeh, S. Aflatoxins in Cereals and Cereal-Based Products: Occurrence, Toxicity, Impact on Human Health, and Their Detoxification and Management Strategies. Toxins 2022, 14, 687. https://doi.org/10.3390/toxins14100687
3. Line 124: Add a phrase “on day 1” after “compared to the control” to make the sentence meaning more clear.
4. In the section of 2.2: there mycotoxins was produced by Fusarium sporotrichioides (NRRL 3299), Fusarium graminearum (NRRL 5883), and Fusarium verticillioides (MRC 826) strain, respectively. In addition to these three types of mycotoxins, it should be analyzed or discussed whether these strains also produce other types of mycotoxins.
